

# The relationship between allergic rhinitis and attention deficit hyperactivity disorder: a systematic review and meta-analysis

Qian Wang[1,2,*], Ruikun Wang[3,*], Mengyao Li[1], Jieqiong Liang[1], Xiaojun Zhan[1], Yingxia Lu[1], Guimin Huang[4] and Qinglong Gu[1,2]

[1] Department of Otolaryngology-Head and Neck Surgery, Capital Institute of Pediatrics, Beijing, China
[2] Graduate School of Peking Union Medical College, Beijing, China
[3] Capital Institute of Pediatrics-Peking University Teaching Hospital, Beijing, China
[4] Child Health Big Data Research Center, Capital Institute of Pediatrics, Beijing, China
* These authors contributed equally to this work.

## ABSTRACT

**Background:** Numerous investigations have examined the potential link between allergic rhinitis (AR) and attention deficit hyperactivity disorder (ADHD). However, some studies show no association between the two diseases. The connection between these two conditions remains inconclusive. This study aimed to conduct a meta-analysis exploring the correlation between AR and ADHD.

**Methods:** We conducted systematic searches of the MEDLINE, EMBASE, Cochrane Library, ERIC, PubMed, Web of Science, and CINAHL databases, up to the year 2023. Subsequently, we conducted a meta-analysis using R 4.2.2, where we computed the pooled odds ratio with a 95% confidence interval to assess the relationship between AR and ADHD within studies exhibiting similar characteristics. Statistical heterogeneity was evaluated by computing the value using the Cochrane Intervention Manual's guidelines. Additionally, subgroup analyses were conducted by stratifying the study population according to gender, age, *etc*. Sensitivity analysis was performed by systematically removing individual studies.

**Results:** In this systematic review, we incorporated 12 eligible studies, collectively encompassing a sample size of 530,360 participants. Within the included studies, heterogeneity was observed, and the utilization of a random-effects model demonstrated a noteworthy correlation between children with ADHD and the presence of AR. Similarly, children with AR exhibited a significant correlation with the occurrence of ADHD. We also found some relationships in subgroup analyses.

**Conclusion:** A substantial correlation is evident between AR and ADHD in children and adolescents. AR may potentially contribute as a risk factor for the onset of ADHD, and conversely, ADHD may heighten the likelihood of developing AR.

Corresponding author
Qinglong Gu, gql71@163.com

## INTRODUCTION

Allergic rhinitis (AR) and attention deficit hyperactivity disorder (ADHD) are two high-prevalence diseases that have significant impacts on children's health. AR significantly affects children's health by exacerbating asthma, contributing to sinusitis, otitis media, and adenoid hypertrophy. AR typically presents with symptoms including rhinorrhea, nasal congestion, and sneezing (*Skoner, 2001*). The symptoms linked to AR can substantially impact the daily routines and the quality of sleep on children, potentially resulting in psychological and behavioral issues (*Yaghmaie, Koudelka & Simpson, 2013*). Approximately 25% of children worldwide are affected by AR (*Skoner, 2001*), and its prevalence has steadily risen in recent years. ADHD is characterized by persistent patterns of inattention and/or hyperactivity-impulsivity that interfere with functioning or development (*Faraone et al., 2003*; *Gau et al., 2005*). According to DSM-5 criteria, these symptoms must be present for at least six months, be inappropriate for the child's developmental level, and significantly impact social, academic, or occupational functioning (*Swanson, Wigal & Lakes, 2009*). The global community prevalence of ADHD in children varies from 2% to 7%, with an average of 5% (*Sayal et al., 2018*). Numerous studies have shown an association between AR and ADHD, indicating that AR could potentially serve as a risk factor for the onset of ADHD (*Chua et al., 2021*; *Schans et al., 2017*; *Yang et al., 2014*), and children with ADHD may be more prone to developing AR (*Brawley et al., 2004*; *Tsai et al., 2011*). *Miyazaki et al. (2017)* also found that children with ADHD are more susceptible to developing AR compared to their peers. However, some research has suggested that there might be no association between AR and ADHD (*Gaitens, Kaplan & Freigang, 1998*), leaving the relationship between the two conditions contentious.

Regarding neurophysiological mechanisms, some studies suggest that interactions between the immune response and the central nervous system (CNS) may contribute to conditions such as autism, impulsive behavior, or hyperactivity in some children (*Chang et al., 2013*; *Comings et al., 2000*; *Lee & Song, 2018*; *Merikangas et al., 2015*; *Miyazaki et al., 2015*; *Petra et al., 2015*). Specifically, allergen exposure triggers interactions between dendritic cells and T cells, leading to the release of inflammatory cytokines. These cytokines can indirectly affect neuronal activity by activating the hypothalamic-pituitary-adrenal axis, influencing neuroimmune pathways (*Steinhoff et al., 2022*; *Bilbo & Schwarz, 2012*; *Kerr et al., 2005*; *Wynne, Henry & Godbout, 2009*). Additionally, cytokines may disrupt neurotransmitter metabolism, such as norepinephrine and dopamine, which are implicated in ADHD (*Chen et al., 2019*). Conversely, ADHD may contribute to AR through elevated stress levels, which trigger neuroimmune responses. Both physical and psychological stressors prompt cytokine production and initiate a cascade of immune reactions. Stress perception by the central nervous system translates into biological responses, activating the hypothalamus and pituitary gland, which then stimulates the autonomic nervous system (*Pondeljak & Lugović-Mihić, 2020*). Similar neuroimmune alterations have been observed in other neurodevelopmental disorders, such as autism and panic disorder (*Brambilla et al., 1994*, *1997*). What's more, both ADHD and AR involve genetic predispositions and environmental triggers, suggesting shared underlying mechanisms. One such mechanism is the signal transducer and activator of transcription 6

(STAT6), which plays a role in immune regulation, cell function, and the pathogenesis of ADHD, further supporting the correlation between AR and ADHD (*Vercelli, 2008*).

The objective of this study is to systematically review the existing evidence, and evaluate and summarize the link between AR and ADHD in children and adolescents through meta-analysis.

Portions of this text were previously published as part of a preprint (https://www.researchsquare.com/article/rs-3926493/v1).

## METHODS

This study adhered to the PRISMA (Preferred Reporting Items for Systematic Reviews and Meta-Analyses) checklist to ensure a comprehensive and transparent reporting of the systematic review and meta-analysis process. The PRISMA checklist guided our approach throughout the study, including the development of the search strategy, study selection criteria, quality assessment procedures, and statistical analysis methods. By following these guidelines, we aimed to enhance the rigor, transparency, and reproducibility of our findings.

### Search strategy

This review protocol was registered with PROSPERO (CRD42022370303). Two researchers performed a systematic literature search across seven databases. Extensive literature research was conducted and literature searches were initiated before registration to identify a clear topic. Four databases were identified: MEDLINE, EMBASE, Cochrane Library, and PubMed. Three additional databases were searched after registration to complete the literature and refine the study: ERIC, Web of Science, and CINAHL. A comprehensive search strategy was developed using a range of keywords, including "Allergic Rhinitis", "Pollen Allergy", "Attention-Deficit Disorder-with-Hyperactivity", "ADHD", and others. The detailed search strategies are listed in Table S1. During the search process, a blend of subject headings and free-text terms was utilized, with logical operators "OR" and "AND" applied. The search was restricted to research conducted in the English language, and the search was current up to 2023. EndNote software version X9 was used to manage the retrieved literature.

### Study selection

During the study eligibility assessment phase, two researchers separately assessed all potentially suitable literature by reviewing titles and abstracts. Titles and abstracts that met the inclusion criteria were chosen for a full-text examination. In cases of discrepancies regarding inclusion, the two researchers will discuss and reach a consensus based on their expertise and access to information, or seek third-party experts' opinions for evaluation, and finally, a professor, QL Gu, will make the final recommendation.

### Inclusion criteria

(1) Cross-sectional, cohort, and case-control studies that categorized patients into either the diagnosed ADHD group and the control group without an ADHD diagnosis or the diagnosed AR group and the control group without an AR diagnosis.

(2) The study participants included children and adolescents aged 18 years and below.

(3) The primary outcome variable was the odds ratio (OR) between ADHD and AR. If OR data were unavailable but calculable, the study was still considered.

(4) Studies were required to include diagnostic methods for both ADHD and AR. If the study provided only scores for ADHD symptoms without a formal diagnosis, consultation with experts was conducted to determine inclusion.

### Exclusion criteria

(1) Studies with insufficient data.

(2) Studies that did not include a control group.

(3) Reviews, guidelines, letters, commentaries, and similar publication types.

(4) Repetitive articles with incomplete data.

### Extraction of data

Data from each eligible article were initially extracted by one researcher using a customized data table. Afterward, another researcher conducted a review to ensure the accuracy of the extracted data. In case of discrepancies during data extraction, consensus was reached through discussion or, if necessary, consultation with a third-party expert. For data extraction, Microsoft Excel from Microsoft Corp. in Redmond, WA, United States, was employed. The following aspects were included:

Basic information: This contained the article's title, author, publication year, and study location.

Study characteristics: This included study type, total number of participants meeting inclusion criteria, their age, and gender. Diagnostic criteria for AR and ADHD specific to each study were recorded.

Study outcomes: This involved ORs, 95% CIs, and other relevant data about the correlation between AR and ADHD. In cases where outcome variables were not explicitly provided, researchers extracted available information from the article and calculated the required study outcome data.

All data were extracted from the manuscripts, and Table 1 summarizes essential information from the included literature.

### Quality assessment

Two researchers independently conducted quality assessments of all eligible studies included in the meta-analysis. In cases where disagreements arose, resolution was achieved through discussions or, if necessary, by consulting with a third-party expert. For cross-sectional studies, we utilized the Risk of Bias Assessment Tool for Nonrandomized Studies (ROBANS), which is recommended by the Agency for Healthcare Research and Quality (AHRQ) (*Huppert et al., 2019*). According to this assessment tool, studies were classified as low quality (0–3 points), moderate quality (4–7 points), or high quality (8–11 points) according to their scores. Inclusion in the meta-analysis was limited to articles scoring above five points, with higher scores indicating higher quality. In the assessment of case-control and cohort studies, we applied the Newcastle-Ottawa Scale

**Table 1 Key features of the literature encompassed in the systematic review.**

| Author, Year | State | Age | Study demographics | Study design | Method assessment attention deficit hyperactivity disorder | Method assessment allergic rhinitis | Quality evaluation |
|---|---|---|---|---|---|---|---|
| Chen et al. (2019) | China Wenzhou | 6–12 | N = 465<br>Males 46.9%<br>Females 53.1% | Cross-sectional study | SNAP-IV + IHS | ARIA + TNSS + SPT + PRQLQ | 5 (AHRQ) |
| Suwan, Akaramethathip & Noipayak (2011) | Thailand | 5–15 | N = 80<br>Males 77.5%<br>Females 22.5% | Case-control study | DSM-IV | SPT | 6 (NOS) |
| Chou et al. (2013) | China Taiwan | 0–17 | N = 221068<br>Males 51.9%<br>Females 48.1% | Cross-sectional study | ICD-9-CM Code 314 | ICD-9-CM Code 477 | 6 (AHRQ) |
| Tsai et al. (2013) | China Taiwan | 7–18 | N = 23460<br>Males 77.9%<br>Females 22.1% | Case-control study | ICD-9-CM Code 314 | ICD-9-CM Code 477 | 6 (NOS) |
| Lee et al. (2014) | Korea | 6–13 | N = 160<br>Males 39.4%<br>Females 60.6% | Case-control study | CBCL + ARS + CAT + DSM-IV | ARIA | 5 (NOS) |
| Kwon et al. (2014) | Korea | 6–9 | N = 4113<br>Males 51.5%<br>Females 48.5% | Case-control study | DSM IV+ADS+ CPRS-HI and CTRS-HI +DuPaul | ISAAC | 8 (NOS) |
| Jameson et al. (2016) | America | 13–18 | N = 7033<br>Males 51.2%<br>Females 48.8% | Cross-sectional study | DSM-IV | Questionnaire | 4 (AHRQ) |
| Nemet et al. (2022) | Israel | 0–18 | N = 132401<br>Males 51.2%<br>Females 48.8% | Cohort study | ICD- 9- CM Code 314+DSM- 5 | ARIA + ICD- 9- CM Code 477 | 8 (NOS) |
| Chang et al. (2019) | China Taiwan | 0–15 | N = 95802<br>Males 50.8%<br>Females 49.2% | Cohort study | ICD-9-CM Code 314 | ICD-9-CM Code 477 | 8 (NOS) |
| Qu et al. (2022) | America | 2–18 | N = 1696<br>Males 49.2%<br>Females 50.8% | Cohort study | ICD-9-CM Code 314 ICD-10-CM Code F90 | ICD-9-CM Code 477 | 7 (NOS) |
| Chen et al. (2013) | China Taiwan | 0–18 | N = 38064<br>Males 62.6%<br>Females 37.4% | Case-control study | ICD-9-CM Code 314 | ICD-9-CM Code 477 | 6 (NOS) |
| Yang, Yang & Wang (2018) | China Taiwan | 3–6 | N = 6018<br>Males 53.3%<br>Females 6.7% | Cross-sectional study | DSM-4 | SPT + ISAAC | 6 (AHRQ) |

**Note:**
Abbreviations: ADS, Attention deficit-hyperactivity disorder diagnostic system; AHRQ, Agency for Healthcare Research and Quality; ARIA, Allergic Rhinitis and its Impact on Asthma; ARS, ADHD Rating Scale; CAT, Comprehensive Attention Test; CBCL, Child Behavior Checklist; CPRS-HI and CTRS-HI, Abbreviated conners parent-teacher rating scale (revised); DSM, Diagnostic and Statistical Manual of Mental Disorders; DuPaul, DuPaul's ADHD rating scales; ICD, International Classification of Diseases; IHS, The inattention/hyperactivity scale; ISAAC, International Study of asthma and allergy in Children; NOS, Newcastle-Ottawa quality assessment scale.; PRQLQ, Pediatric Rhino conjunctivitis quality of life questionnaire; SNAP-IV, The Swanson, Nolan, and Pelham version IV scale; SPT, Skin prick test; TNSS, The Total Nasal Symptoms Score.

(NOS) (*Lo, Mertz & Loeb, 2014*). Based on NOS scoring criteria, studies were classified as low quality (<4 points), moderate quality (4–6 points), or high quality (7–9 points). Again, inclusion in the meta-analysis was limited to studies scoring above five points, with higher scores indicating higher quality. This quality assessment approach enabled us to determine the level of quality among studies included in the meta-analysis to ensure the reliability and credibility of the obtained results. For details, please see Table S2.

## Statistical methods

All data analyses were performed using R 4.2.2 software. Two researchers independently analyzed the data, with the primary outcome of this analysis focusing on the OR for AR prevalence in ADHD *vs*. controls, as well as the OR for ADHD prevalence in AR *vs*. controls. Due to the substantial potential for between-study variance arising from different study designs and populations, all data were computed with 95% CIs using a random-effects model. To evaluate statistical heterogeneity among the summarized data, we employed both the $I^2$ statistic and the chi-square-based $Q$ statistic. As per the Cochran's Handbook for systematic reviews of interventions, the $I^2$ value can be interpreted as follows: When the $I^2$ value falls between 0% to 40%, it may indicate that the heterogeneity is not of substantial importance. In cases where the $I^2$ value ranges from 40% to 75%, it represents a moderate level of heterogeneity. An $I^2$ value within the range of 75% to 100% indicates a considerable level of heterogeneity.

Moreover, with the intention of pinpointing significant sources of heterogeneity that must not be overlooked, we conducted subgroup analyses. These analyses were grounded in various characteristics of the study population, including gender, age, study type, study quality, AR diagnostic methods, and region. Subsequently, we combined effect sizes were calculated. We performed a sensitivity analysis to evaluate the robustness of the results by sequentially excluding individual studies and assessing their effects on the final effect estimate. This process helps ensure the reliability and stability of the meta-analysis results. A significance level of $p < 0.05$ was used to indicate the presence of statistical significance. The assessment of publication bias was conducted using the Egger linear regression test. This test helps identify any potential bias in the publication of studies and contributes to the overall assessment of the reliability of the meta-analysis results.

# RESULTS

## Inclusion of studies

In total, our comprehensive search across seven databases using the respective search strategies initially identified 1,183 articles. After eliminating duplicate articles, 802 unique articles remained for further evaluation. Subsequently, 517 articles were excluded based on irrelevant content after reviewing titles and abstracts, leaving 285 articles. After a full-text review of these 285 articles, 83 reviews, guidelines, letters, conference reports, case reports, and short-term surveys were excluded. Additionally, 103 articles were excluded due to the unavailability of original data or the absence of the required outcome variables. Three articles were excluded because of data duplication and small sample sizes. Ultimately, 12 articles satisfied the criteria for inclusion in the systematic review and meta-analysis
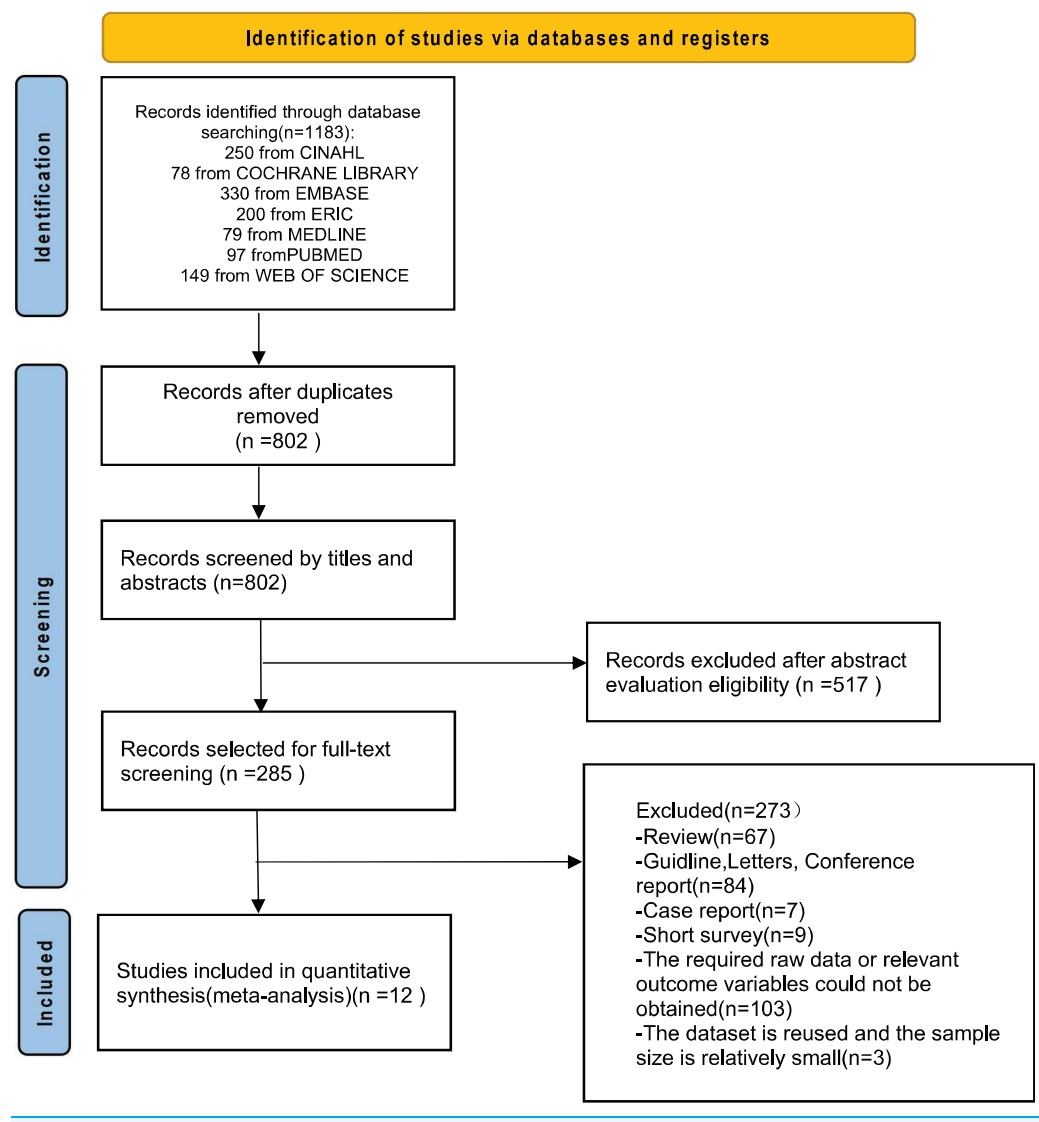

**Figure 1 Flowchart depicting the process of study selection.**

(*Chang et al., 2019*; *Chen et al., 2019*, *2013*; *Chou et al., 2013*; *Jameson et al., 2016*; *Kwon et al., 2014*; *Lee et al., 2014*; *Nemet et al., 2022*; *Qu et al., 2022*; *Suwan, Akaramethathip & Noipayak, 2011*; *Tsai et al., 2013*; *Yang, Yang & Wang, 2018*). These eligible studies comprised five case-control studies, four cross-sectional studies, and three cohort studies. These studies originated from various regions, including China, South Korea, Thailand, Israel, and the United States. A summary of the study selection process is presented in Fig. 1.

## Synthesis of results

We conducted separate meta-analyses using two studies reporting the prevalence of ADHD in AR/non-AR patients and ten studies reporting the prevalence of AR in ADHD/non-ADHD patients. After excluding two studies investigating the relationship

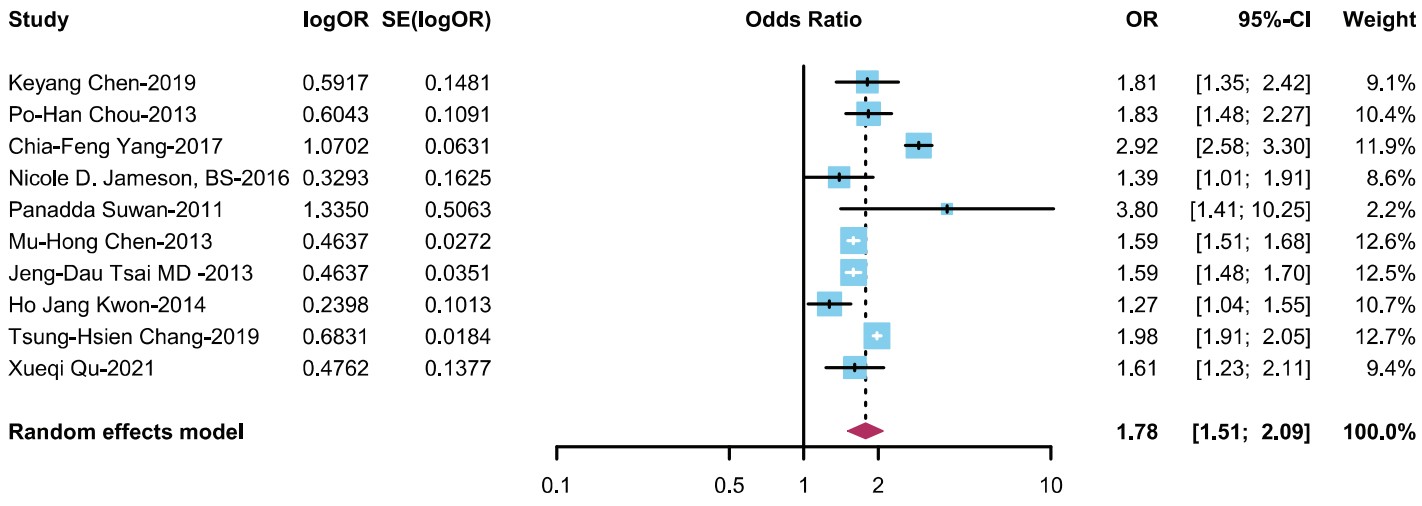

| Study | logOR | SE(logOR) | Odds Ratio | OR | 95%-CI | Weight |
|---|---|---|---|---|---|---|
| Keyang Chen-2019 | 0.5917 | 0.1481 | | 1.81 | [1.35; 2.42] | 9.1% |
| Po-Han Chou-2013 | 0.6043 | 0.1091 | | 1.83 | [1.48; 2.27] | 10.4% |
| Chia-Feng Yang-2017 | 1.0702 | 0.0631 | | 2.92 | [2.58; 3.30] | 11.9% |
| Nicole D. Jameson, BS-2016 | 0.3293 | 0.1625 | | 1.39 | [1.01; 1.91] | 8.6% |
| Panadda Suwan-2011 | 1.3350 | 0.5063 | | 3.80 | [1.41; 10.25] | 2.2% |
| Mu-Hong Chen-2013 | 0.4637 | 0.0272 | | 1.59 | [1.51; 1.68] | 12.6% |
| Jeng-Dau Tsai MD -2013 | 0.4637 | 0.0351 | | 1.59 | [1.48; 1.70] | 12.5% |
| Ho Jang Kwon-2014 | 0.2398 | 0.1013 | | 1.27 | [1.04; 1.55] | 10.7% |
| Tsung-Hsien Chang-2019 | 0.6831 | 0.0184 | | 1.98 | [1.91; 2.05] | 12.7% |
| Xueqi Qu-2021 | 0.4762 | 0.1377 | | 1.61 | [1.23; 2.11] | 9.4% |
| **Random effects model** | | | | 1.78 | [1.51; 2.09] | 100.0% |

Heterogeneity: $I^2 = 93\%$, $\tau^2 = 0.0541$, $p < 0.01$

**Figure 2** Forest plot illustrating the association between the occurrence of AR in ADHD patients (*Chen et al., 2019*; *Chou et al., 2013*; *Yang, Yang & Wang, 2018*; *Jameson et al., 2016*; *Suwan, Akaramethathip & Noipayak, 2011*; *Chen et al., 2013*; *Tsai et al., 2013*; *Kwon et al., 2014*; *Chang et al., 2019*; *Qu et al., 2022*).

between AR/non-AR patients and ADHD, we proceeded with subgroup analysis and sensitivity analysis using the remaining ten studies.

## Relationship between ADHD/Non-ADHD patients and AR

A total of 10 studies (*Chang et al., 2019*; *Chen et al., 2019, 2013*; *Chou et al., 2013*; *Kwon et al., 2014*; *Qu et al., 2022*; *Suwan, Akaramethathip & Noipayak, 2011*; *Tsai et al., 2013*; *Yang, Yang & Wang, 2018*) described the incidence/prevalence of AR in patients with ADHD and non-ADHD patients, with a cumulative sample size of $n = 397,799$. The meta-analysis results revealed that the risk of developing AR in the ADHD group was 1.84 times greater than that in the control group (OR: 1.84, 95% CI [1.55–2.19]) (Fig. 2).

## Relationship between AR/Non-AR patients and ADHD

Two studies (*Nemet et al., 2022*; *Yang, Yang & Wang, 2018*) described the prevalence of ADHD in patients with AR and non-AR patients, with a combined sample size of $n = 132,561$. The meta-analysis results indicated that the likelihood of developing ADHD in the AR group was 3.96 times greater than that in the control group (OR: 3.96, 95% CI [3.80–4.12]) (Fig. 3).

## Subgroup analysis

### The relationship between ADHD and AR is categorized by gender

To explore the influence of gender on the relationship between ADHD and AR, we conducted subgroup analysis on three studies with gender-matched data (*Chen et al., 2019*; *Chou et al., 2013*; *Tsai et al., 2013*) to explore the influence of different genders on the connection between children with ADHD and prevalence of AR. The results showed (Fig. 4) an OR of 1.42 (95% CI [1.24–1.6]) in the male group and an OR of 1.97 (95% CI [1.52–2.56]) in the female group.

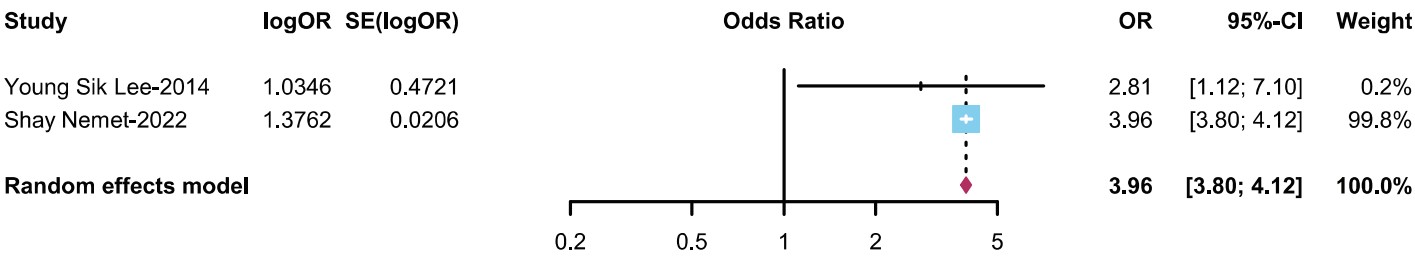

| Study | logOR | SE(logOR) | Odds Ratio | OR | 95%-CI | Weight |
|---|---|---|---|---|---|---|
| Young Sik Lee-2014 | 1.0346 | 0.4721 | | 2.81 | [1.12; 7.10] | 0.2% |
| Shay Nemet-2022 | 1.3762 | 0.0206 | | 3.96 | [3.80; 4.12] | 99.8% |
| **Random effects model** | | | | **3.96** | **[3.80; 4.12]** | **100.0%** |

Heterogeneity: $I^2 = 0\%$, $\tau^2 = 0$, $p = 0.47$

**Figure 3** Forest plot illustrating the association between the occurrence of ADHD in AR patients (*Nemet et al., 2022*; *Lee et al., 2014*).

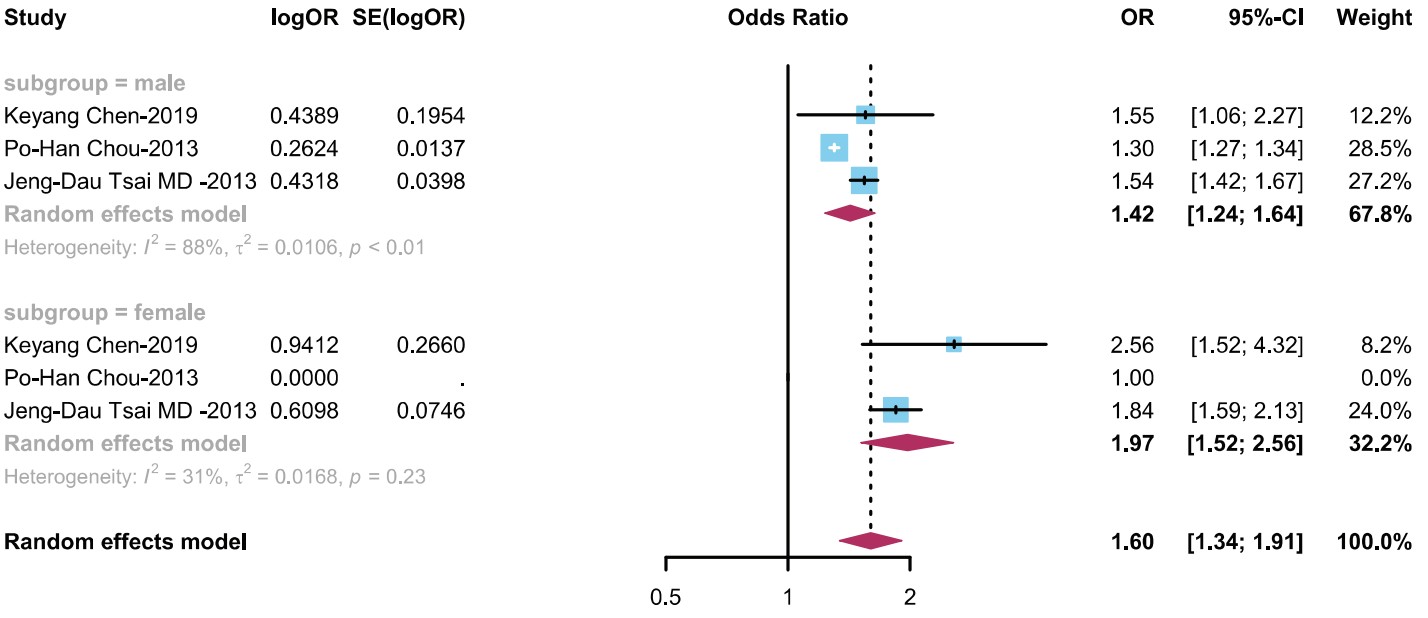

| Study | logOR | SE(logOR) | Odds Ratio | OR | 95%-CI | Weight |
|---|---|---|---|---|---|---|
| **subgroup = male** | | | | | | |
| Keyang Chen-2019 | 0.4389 | 0.1954 | | 1.55 | [1.06; 2.27] | 12.2% |
| Po-Han Chou-2013 | 0.2624 | 0.0137 | | 1.30 | [1.27; 1.34] | 28.5% |
| Jeng-Dau Tsai MD -2013 | 0.4318 | 0.0398 | | 1.54 | [1.42; 1.67] | 27.2% |
| **Random effects model** | | | | **1.42** | **[1.24; 1.64]** | **67.8%** |
| Heterogeneity: $I^2 = 88\%$, $\tau^2 = 0.0106$, $p < 0.01$ | | | | | | |
| **subgroup = female** | | | | | | |
| Keyang Chen-2019 | 0.9412 | 0.2660 | | 2.56 | [1.52; 4.32] | 8.2% |
| Po-Han Chou-2013 | 0.0000 | . | | 1.00 | | 0.0% |
| Jeng-Dau Tsai MD -2013 | 0.6098 | 0.0746 | | 1.84 | [1.59; 2.13] | 24.0% |
| **Random effects model** | | | | **1.97** | **[1.52; 2.56]** | **32.2%** |
| Heterogeneity: $I^2 = 31\%$, $\tau^2 = 0.0168$, $p = 0.23$ | | | | | | |
| **Random effects model** | | | | **1.60** | **[1.34; 1.91]** | **100.0%** |

Heterogeneity: $I^2 = 90\%$, $\tau^2 = 0.0283$, $p < 0.01$
Test for subgroup differences: $\chi_1^2 = 4.61$, df = 1 ($p = 0.03$)

**Figure 4** Forest plot depicting the relationship between the occurrence of AR in male and female ADHD patients (*Chen et al., 2019*; *Chou et al., 2013*; *Tsai et al., 2013*).

### The relationship between ADHD and AR is categorized by age

To assess how age impacts the association between ADHD and AR, we conducted a subgroup analysis on seven studies (*Chang et al., 2019*; *Chen et al., 2019*, *2013*; *Jameson et al., 2016*; *Kwon et al., 2014*; *Suwan, Akaramethathip & Noipayak, 2011*; *Tsai et al., 2013*) with age stratification data. This analysis explored the impact of different age groups on the relationship between children with ADHD and the prevalence of AR. Patients were divided into age groups of 6–9 years, 9–15 years, and over 15–18 years based on average age. The results showed (Fig. 5) an OR of 1.45 (95% CI [1.17–1.80]) in the 6–9 years group, an OR of 2.24 (95% CI [1.16–4.33]) in the 9–15 years group, and OR of 1.94 (95% CI [1.52–2.47]) in the 15–18 years group.

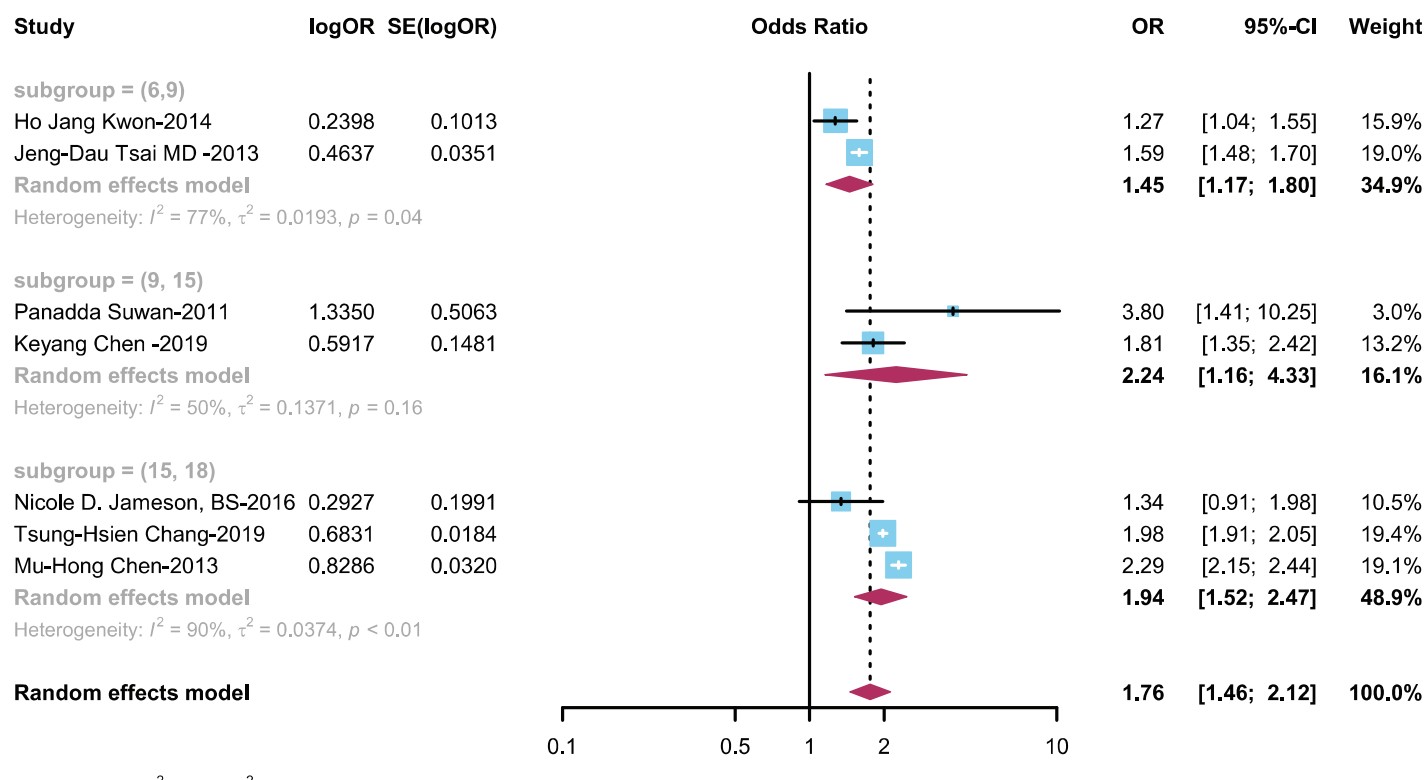

**Figure 5** Forest plot illustrating the relationship between the occurrence of AR in ADHD patients across various age groups (*Kwon et al., 2014*; *Tsai et al., 2013*; *Suwan, Akaramethathip & Noipayak, 2011*; *Chen et al., 2019*; *Jameson et al., 2016*; *Chen et al., 2013*; *Chang et al., 2019*).

### The relationship between ADHD and AR is categorized by study type

To investigate the influence of different study types on the link between ADHD and AR, we conducted a subgroup analysis on ten studies (*Chang et al., 2019*; *Chen et al., 2019*, *2013*; *Chou et al., 2013*; *Jameson et al., 2016*; *Kwon et al., 2014*; *Qu et al., 2022*; *Suwan, Akaramethathip & Noipayak, 2011*; *Tsai et al., 2013*; *Yang, Yang & Wang, 2018*) stratified by case-control, cohort, and cross-sectional study types. This analysis was conducted with the aim of evaluating how different study types influence the relationship between children with ADHD and the prevalence of AR. The results showed (Fig. S1) an OR of 1.80 (95% CI [1.28–2.53]) in the case-control study group, an OR of 1.87 (95% CI [1.56–2.24]) in the cohort study group, and an OR of 1.95 (95% CI [1.43–2.68]) in the cross-sectional study group.

### The relationship between ADHD and AR is categorized by study quality

To examine how different levels of study quality influence the connection between ADHD and AR, we conducted a subgroup analysis on ten studies (*Chang et al., 2019*; *Chen et al., 2019*, *2013*; *Chou et al., 2013*; *Jameson et al., 2016*; *Kwon et al., 2014*; *Qu et al., 2022*; *Suwan, Akaramethathip & Noipayak, 2011*; *Tsai et al., 2013*; *Yang, Yang & Wang, 2018*) stratified by high and moderate quality. This analysis was conducted with the objective of evaluating

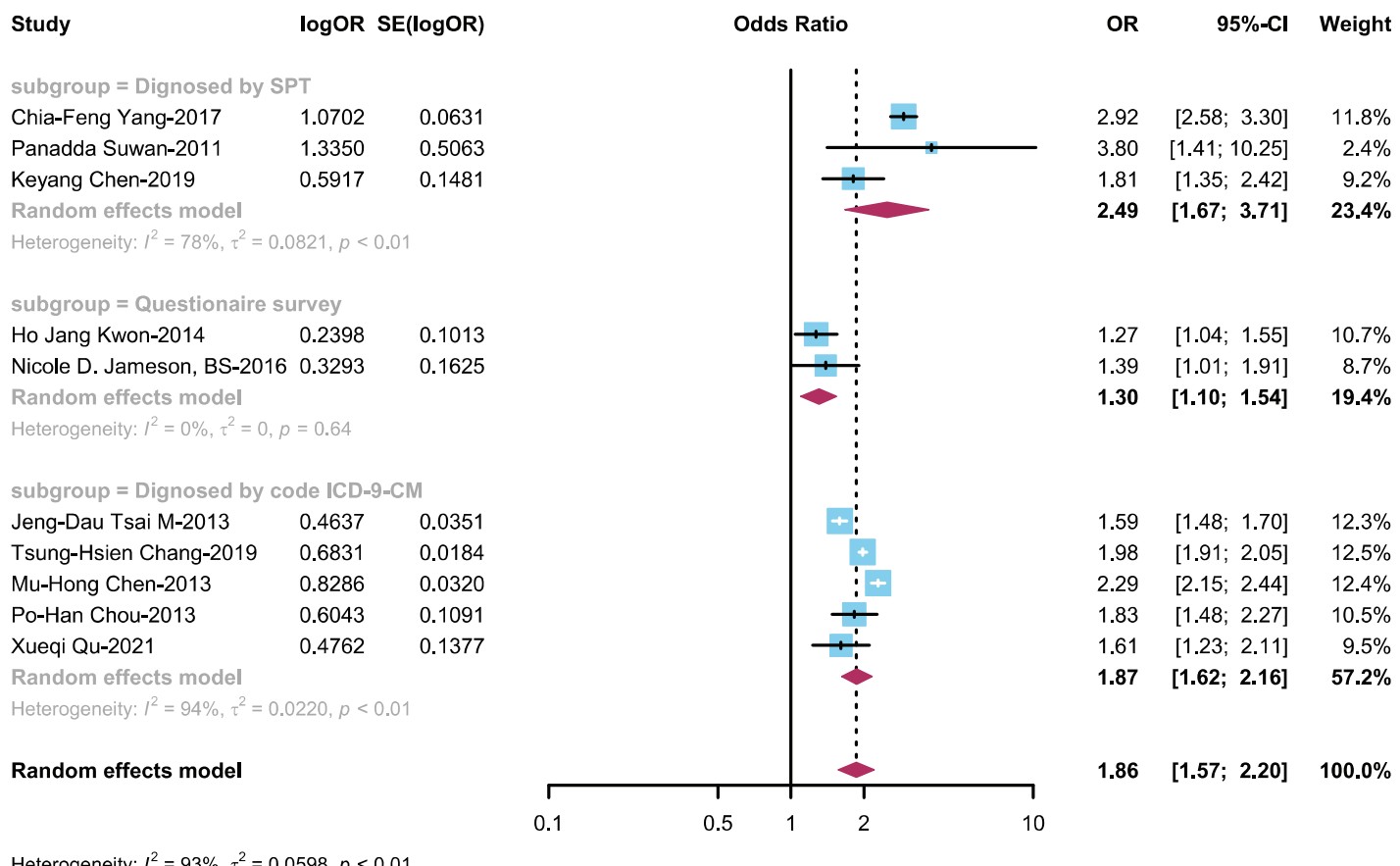

**Figure 6** Forest plot illustrating the connection between the occurrence of AR in ADHD patient based on diagnostic method (*Yang, Yang & Wang, 2018*; *Suwan, Akaramethathip & Noipayak, 2011*; *Chen et al., 2019*; *Kwon et al., 2014*; *Jameson et al., 2016*; *Tsai et al., 2013*; *Chang et al., 2019*; *Chen et al., 2013*; *Chou et al., 2013*; *Qu et al., 2022*).

how varying levels of study quality impact the link between children with ADHD and the prevalence of AR. The results showed (Fig. S2) an OR of 1.62 (95% CI [1.24–2.12]) in the moderate-quality studies group and an OR of 1.97 (95% CI [1.58–2.45]) in the high-quality studies group.

**The relationship between ADHD and AR is categorized by diagnostic method**

To explore the influence of different diagnostic methods for AR on the ADHD-AR association, we conducted a subgroup analysis on ten studies (*Chang et al., 2019*; *Chen et al., 2019*, *2013*; *Chou et al., 2013*; *Jameson et al., 2016*; *Kwon et al., 2014*; *Qu et al., 2022*; *Suwan, Akaramethathip & Noipayak, 2011*; *Tsai et al., 2013*; *Yang, Yang & Wang, 2018*) stratified by various AR diagnostic methods, including the SPT (skin prick test), questionnaire survey only, and ICD-9-CM. This analysis was performed with the purpose of evaluating how different AR diagnostic methods affect the relationship between children with ADHD and the development of AR. The results showed (Fig. 6) an OR of 2.49 (95% CI [1.67–3.71]) in the SPT diagnosis group, an OR of 1.30 (95% CI [1.10–1.54]) in the

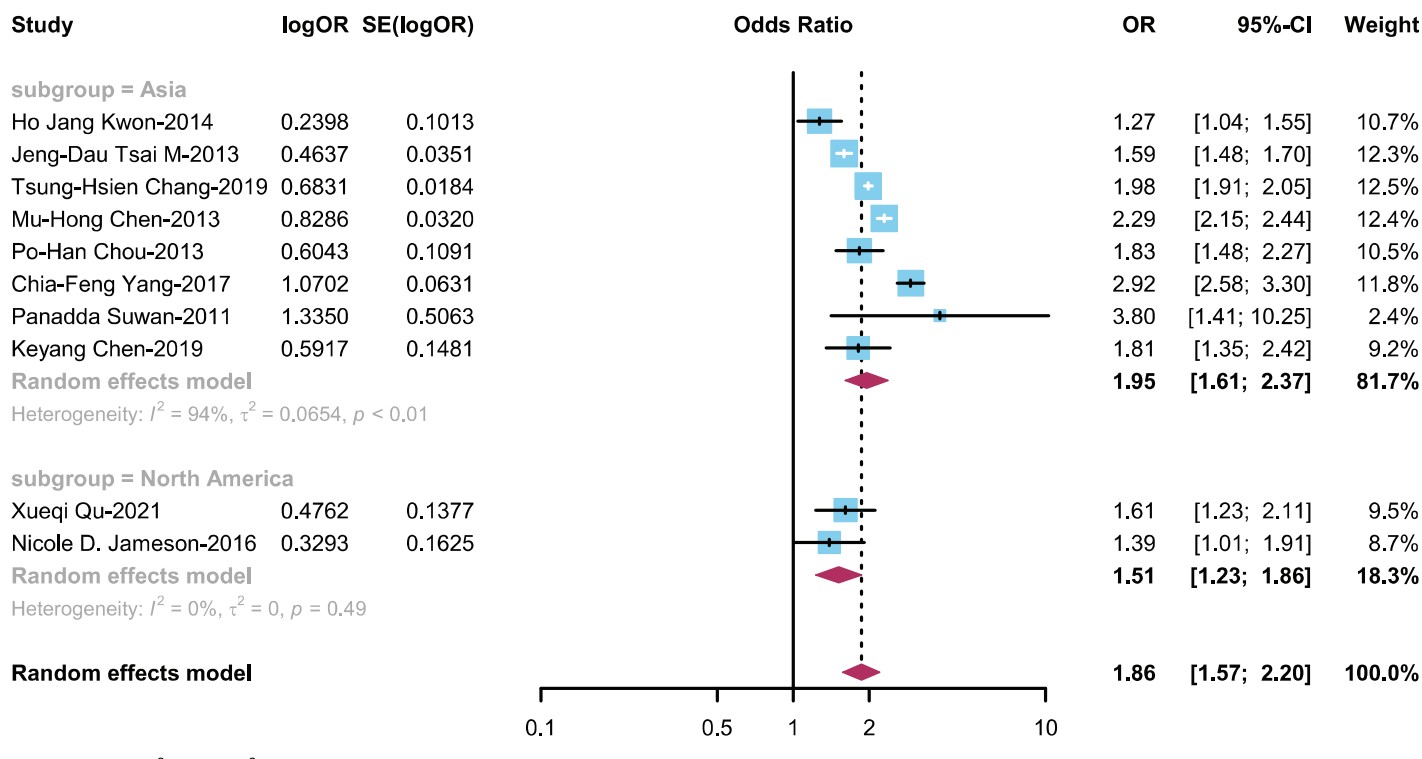

**Figure 7** Forest plot illustrating the connection between the occurrence of AR in ADHD patient based on regions (*Kwon et al., 2014*; *Tsai et al., 2013*; *Chang et al., 2019*; *Chen et al., 2013*; *Chou et al., 2013*; *Yang, Yang & Wang, 2018*; *Suwan, Akaramethathip & Noipayak, 2011*; *Chen et al., 2019*; *Qu et al., 2022*; *Jameson et al., 2016*).

questionnaire survey-only diagnosis group, and OR of 1.87 (95% CI [1.62–2.17]) in the ICD-9-CM diagnosis group.

### The relationship between ADHD and AR is categorized by region

To investigate how study regions influence the connection between ADHD and AR, we performed a subgroup analysis on 10 studies (*Chang et al., 2019*; *Chen et al., 2019*, *2013*; *Chou, 2013*; *Jameson et al., 2016*; *Kwon et al., 2014*; *Qu et al., 2022*; *Suwan, Akaramethathip & Noipayak, 2011*; *Tsai et al., 2013*; *Yang, Yang & Wang, 2018*) stratified by different research regions, distinguishing between Asian and North American regions. This analysis was carried out to explore how different research regions impact the link between children with ADHD and the prevalence of AR. The OR for the Asian region group was 1.95 (95% CI [1.61–2.37]), while the OR for the North American region group was 1.51 (95% CI [1.23–1.86]) (Fig. 7).

### Risk of bias

Quality assessments were conducted for the 12 included studies using relevant criteria. Among the four cross-sectional studies, four were rated as moderate-quality studies. In the case-control studies, four were assessed as having moderate to low risk, and one was assessed as low risk. Among the cohort studies, three were assessed as low risk. (Table S2).

## Publication bias

The results of Egger's test suggested a low likelihood of publication bias ($t = -0.61$, $p = 0.5565$).

## Sensitivity analysis

Sensitivity analyses were performed to assess the stability of the study results. Firstly, we performed sensitivity analysis by sequentially excluding individual studies, and the resulting pooled odds ratio was 1.84 (95% CI [1.55–2.19]). It was observed that all studies were relatively evenly distributed on both sides of the vertical line (Fig. S3), suggesting low sensitivity in this study and relatively stable results. Secondly, we conducted sensitivity analysis by excluding four cross-sectional studies based on the literature study type. After removing these studies, the combined OR for the meta-analysis was 1.78 (95% CI [1.46–2.18]), the conclusion remained in line with the previous results, underscoring that the outcomes of the meta-analysis remained relatively consistent even after excluding studies with a risk of bias (Fig. S4).

## DISCUSSION

This systematic review and meta-analysis investigated the link between allergic rhinitis (AR) and ADHD across 12 studies involving a total of 530,360 participants. Our primary finding indicated a significant correlation between AR and ADHD. Furthermore, subgroup analyses, considering various demographic variables, revealed that female individuals with ADHD were more prone to developing AR compared to their male counterparts. Additionally, the risk of AR was higher among individuals with ADHD aged 9–15 years in various age groups. Interestingly, we observed no significant distinctions across different study types, but the relationship between ADHD and AR was notably more pronounced in studies of higher quality. Remarkably, when we examined the data by categorizing participants based on their geographical regions, it became evident that the association between ADHD and AR was notably stronger in Asia compared to other regions.

Moreover, various studies have concentrated on assessing the influence of AR on children's mental health and behavior, with particular attention to the connection between AR and ADHD as a pivotal area of interest. Multiple studies have already found a strong correlation between AR and ADHD (*Lin et al., 2016*; *Shyu et al., 2012*; *Suwan, Akaramethathip & Noipayak, 2011*). Some studies suggest that treating AR can significantly reduce the severity of ADHD (*Buske-Kirschbaum et al., 2013*; *Yang et al., 2016*). Conversely, ADHD has been examined for its impact on AR, such as ADHD treatment improving allergic symptoms (*van der Schans et al., 2020*). This suggests that the relationship between AR and ADHD is not a simple one-way influence but rather a mutual interaction. However, some studies indicate that there is no significant correlation between neurodevelopmental disorders and allergic reactions, suggesting that there may not be a clear link between ADHD and AR (*Blank & Remschmidt, 1994*), indicating the need for further investigation into the complex interplay between ADHD and AR.

## Subgroup analysis

Further analysis delved into the nuanced dynamics of the association between AR and ADHD across various subgroups, including gender, age, study quality, diagnostic methods, and geographic region.

Recognizing the divergent prevalence rates of ADHD and AR among different genders, our analysis indicates a notable gender-specific association. Specifically, we observed a heightened association between ADHD and AR in females (OR = 1.97; 95% CI [1.52–2.56]). This suggests that gender-specific factors, such as hormonal influences and differences in immune responses, may contribute to the co-occurrence of these conditions. The role of dopamine systems, which have been implicated in both ADHD and AR, may also vary by gender, as hormonal fluctuations and genetic factors can modulate dopaminergic pathways differently in males and females (*Epperson et al., 2015*; *Fröhlich et al., 2017*; *Leffler, Stumbles & Strickland, 2018*; *Willcutt, 2012*). To provide a clearer understanding, future research should focus on these gender-specific mechanisms and explore how they might influence the prevalence and severity of ADHD and AR. This will not only enhance our knowledge of these conditions but also inform gender-specific interventions in clinical practice.

Moreover, considering the differential progression of ADHD and AR across age groups, we explored their relationship with age dynamics. Notably, a more pronounced association between ADHD and AR was observed in the 9–15 years age bracket (OR = 2.24; 95% CI [1.16–4.33]). The heightened association between ADHD and AR among individuals aged 9–15 years highlights the importance of age-specific susceptibility factors in the manifestation of these conditions. This finding suggests that during this critical developmental period, individuals may be more vulnerable to the overlapping pathophysiological mechanisms underlying ADHD and AR. It also underscores the significance of early detection and intervention strategies targeting both conditions, particularly during adolescence when symptoms may exacerbate and impact daily functioning and quality of life. Further research is warranted to elucidate the specific biological and environmental factors contributing to this age-related vulnerability and to develop tailored interventions for this age group.

Across various study designs, including cohort, case-control, and cross-sectional studies, evidence consistently supported an association between ADHD and AR. Notably, studies with more rigorous methodologies showed a more pronounced effect size (OR = 1.97; 95% CI [1.58–2.45]), suggesting that the close relationship between ADHD and AR becomes more evident when methodological rigor is applied. This finding indicates that the stronger associations observed in high-quality studies likely reflect the true biological and clinical relationship between these conditions. Therefore, these results emphasize the importance of using robust methodologies in future research to accurately capture the complex interplay between ADHD and AR, ultimately advancing our understanding and informing clinical practice.

Furthermore, considering the influence of diagnostic methods on disease prevalence, we stratified by AR diagnostic approaches, revealing that studies employing the skin prick test

(SPT) method demonstrated a stronger association between ADHD and AR (OR = 2.49; 95% CI [1.67–3.71]). The SPT method is recognized as the gold standard for AR diagnosis (*Nelson et al., 1993*), likely contributes to more accurate assessments and higher diagnostic rates (*Onore, Careaga & Ashwood, 2012*; *Wynne, Henry & Godbout, 2009*), thereby amplifying the observed association. This emphasizes the critical role of diagnostic accuracy in elucidating the relationship between these two conditions and underscores the importance of employing standardized diagnostic procedures in future research endeavors.

Both ADHD and AR entail genetic-environment interactions in their pathogenesis, suggesting shared underlying genetic mechanisms (*Chen et al., 2019*). To explore the association between ADHD and AR, we stratified by region. Notably, a robust connection between ADHD and AR was evident in the Asian region (OR = 1.95; 95% CI [1.61–2.37]), highlighting potential regional variations in disease susceptibility and contributing factors. These findings suggest that regional differences in environmental exposures and other factors may influence the manifestation and interplay of ADHD and AR. Further research into the specific environmental determinants within different regions could provide valuable insights into the complex etiology of these conditions and guide targeted interventions and preventive measures.

Overall, these subgroup analyses enhance our understanding of the complex interplay between ADHD and AR, highlighting the importance of considering demographic, methodological, and regional factors in elucidating their association. These findings have implications for clinical practice, public health interventions, and future research aimed at mitigating the burden of comorbid ADHD and AR.

## Strengths and weaknesses

The study is notable for some reasons: (1) Comprehensive analysis: This study employed systematic review and meta-analysis methods, integrating data from 12 studies covering a large sample of participants, thoroughly investigating the association between AR and ADHD. (2) Subgroup analysis: By considering various demographic variables such as gender, age, study quality, diagnostic methods, and geographic region, subgroup analyses were conducted, providing insightful variations in the association among different populations. (3) Quality assessment: The inclusion of studies with varying quality levels allowed for a more comprehensive evaluation of the relationship between ADHD and AR, emphasizing the impact of study rigor on the observed association. (4) Bidirectional influence: Acknowledgment of the bidirectional relationship between AR and ADHD in the discussion provides direction for future research.

This study also has certain limitations: (1) Limited causality: Due to the inclusion of observational studies, this research can only establish an association between AR and ADHD, rather than determining causality or underlying pathophysiological mechanisms. (2) Selection bias: The inclusion criteria of the study may introduce selection bias, potentially limiting the representativeness of the results, especially concerning specific geographic regions and populations. (3) Generalizability: Limited diversity in the study populations and dominance of certain geographic regions may restrict the generalizability of the findings, necessitating more representative studies with diverse samples.

(4) Methodological variability: Differences in the diagnostic methods for AR and ADHD across studies may impact the observed association, highlighting the need for standardized diagnostic methods in future research.

In this systematic review, we performed a meta-analysis based on systematic evaluation and combined previous research results, resulting in a relatively large sample size and confirming the link between AR and ADHD. This meta-analysis also explored more precise connections between ADHD and AR by considering gender disparities, variances in age, distinctions in study types and their quality, diverse diagnostic approaches for AR, and regional variations across the studies. The results of this investigation need further validation and exploration. Despite its limitations, the study results still provide preliminary clues about the potential connection between AR and ADHD. Further understanding of the etiology of ADHD/AR may reveal the underlying pathogenic mechanisms for preventing and/or treating these conditions in children. Our analysis offers important guidance and insights for future research.

## CONCLUSIONS

Our research findings confirm a significant association between AR and ADHD, suggesting that clinical practitioners should consider preventing or timely detecting the occurrence of ADHD when managing children with AR, and vice versa. However, the causal relationship and underlying mechanisms of this association remain unclear and require further investigation.

### Funding

This study was supported by the Key Program of Capital's Funds for Health Improvement and Research (2022-1-2101) and the Beijing Municipal Natural Science Foundation (7232010). The funders had no role in study design, data collection and analysis, decision to publish, or preparation of the manuscript.

### Grant Disclosures

The following grant information was disclosed by the authors:
Key Program of Capital's Funds for Health Improvement and Research: 2022-1-2101.
Beijing Municipal Natural Science Foundation: 7232010.

### Competing Interests

The authors declare that they have no competing interests.

### Author Contributions

- Qian Wang conceived and designed the experiments, performed the experiments, analyzed the data, prepared figures and/or tables, authored or reviewed drafts of the article, and approved the final draft.
- Ruikun Wang conceived and designed the experiments, performed the experiments, analyzed the data, prepared figures and/or tables, and approved the final draft.

- Mengyao Li conceived and designed the experiments, performed the experiments, prepared figures and/or tables, and approved the final draft.
- Jieqiong Liang conceived and designed the experiments, prepared figures and/or tables, and approved the final draft.
- Xiaojun Zhan conceived and designed the experiments, prepared figures and/or tables, authored or reviewed drafts of the article, and approved the final draft.
- Yingxia Lu conceived and designed the experiments, prepared figures and/or tables, authored or reviewed drafts of the article, and approved the final draft.
- Guimin Huang conceived and designed the experiments, analyzed the data, prepared figures and/or tables, authored or reviewed drafts of the article, and approved the final draft.
- Qinglong Gu conceived and designed the experiments, performed the experiments, analyzed the data, prepared figures and/or tables, authored or reviewed drafts of the article, and approved the final draft.

## Data Availability

This is a systematic review/meta-analysis.

## Supplemental Information

Supplemental information for this article can be found online at http://dx.doi.org/10.7717/peerj.18287#supplemental-information.

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
