# Peer review of "The relationship between allergic rhinitis and attention deficit hyperactivity disorder: a systematic review and meta-analysis"

_PeerJ, doi:10.7717/peerj.18287_

## Round 0.1 · original submission · Minor Revisions

Both reviewers have provided only minor comments which you must address. Reviewer 2 has noted that there are many other similar studies, which means you should highlight where/how your findings add to the existing literature.

Reviewer 1 ·

Basic reporting

In the introduction section, lines 56-57 states: 'ADHD is typified by the presence of attention deficits and/or hyperactivity that are considered developmentally inappropriate. This neurobehavioral disorder is prevalent among children.' To enhance accuracy, it is important to fully describe the criteria for ADHD, most commonly outlined by the DSM-5. ADHD is characterized by persistent patterns of inattention and/or hyperactivity-impulsivity that interfere with functioning or development. According to DSM-5 criteria, these symptoms must be present for at least six months, be inappropriate for the child's developmental level, and significantly impair social, academic, or occupational functioning. It is also crucial to note that some children may exhibit inappropriate behaviors for their age. Still, if these behaviors do not affect their functional abilities, they are not classified as ADHD.

Experimental design

No comment

Validity of the findings

- Children with poorly controlled AR can experience sleep problems such as snoring or OSA. These sleep issues can lead to inattentiveness, potentially exacerbating ADHD symptoms. Additionally, itching symptoms from AR may manifest as fidgeting, which can be misinterpreted as hyperactivity. It is crucial to conduct a subgroup analysis of AR complications in children with ADHD to determine if these complications are related to the severity and control of AR symptoms or if they stem solely from AR.

Additional comments

No comment

·

Basic reporting

1. The manuscript is well written, employing proper English; however, some structural issues remain. For instance, the first paragraph of the introduction section needs to be more cohesive; clinical background and its impact on patients’ daily functioning should be reported adjacent to each other, followed by epidemiologic reports.
2. The background section lacks sufficient information about ADHD and its symptoms. More detail is also required to briefly discuss the possible mechanisms of association between ADHD and AR
3. More similar studies, such as the work of Miyazaki et al. (2017), should be referenced.

4. Overall, the manuscript's formatting adheres to Cochrane Intervention Manual guidelines.
5. It is unnecessary to report authors’ specific contributions in the methods section. This belongs to the “authors’ contribution” section at the end of the article
6. A more comprehensive discussion is needed to clarify the relationship between ADHD association with AR and its relation to gender. For example, the reference about dopamine systems in line 317 seems a bit out the context and should be further supported by additional sources.
7. In the discussion section, the paragraph comparing the accuracy and quality of the included studies appears overly obvious; it does not contribute new information. This could be revised to indicate that the stronger association between ADHD and AR in more rigorous studies reflects the close association of these conditions. The current discussion lacks noteworthy insights.
8. In line 362 “To explore the association between ADHD and AR in varying racial and environmental contexts”, the racial variance is not evaluated in this study and should not be mentioned.
9. Some information presented in the “Pathophysiological Mechanism” section does not corelate with the objective of the study. for instance, allergen exposure, stress related mechanisms, and neuroimmune pathways are not assessed as variables or cofounder factors in this study. The pathophysiological mechanisms debated in the discussion section should be in-line and explanatory of the results. The current “Pathophysiological Mechanism” section would be more appropriately placed in the introduction.
10. Please mention the use of PRISMA checklist in the methods section.

Experimental design

1. This study falls well within the scope of this journal.
2. Given the extensive data and previous meta-analyses, the necessity of this study is questionable. It could fill more gaps in our understanding of the association of ADHD and AR, if the researchers evaluated other factors such as the role of interventions in reducing the burden of these conditions or the risk factors of their concomitance.
3. The methods are adequately described with sufficient detail, and the technical aspects of the methodology are appropriate

Validity of the findings

1. This study reviews the association of ADHD and AR in multiple diverse studies, which shed light on the concomitance of these conditions; nevertheless, the result and objective of the study lacks novelty. The researchers could encompass more variables (such as risk factors, association with other conditions, and other relevant factors) to provide a deeper understanding of the matter.
2. The underlying data is well presented and statistically acceptable. The discussion section requires major revision, so that the background information used in the article explain the results more effectively.

---

## Round 0.2 · accepted · Accept

I have confirmed that the authors have addressed all of the reviewers' comments.

Reviewer 1 ·

Basic reporting

No comment

Experimental design

No comment

Validity of the findings

No comment

Additional comments

No comment

·

Basic reporting

1. The overall English structure of the manuscript is improved.
2. The introduction section is now more informative and properly sets a background for the research.
3. The method section is revised properly, and excessive information is removed.
4. The discussion section is now well organized and better reflects the objectives of the study.

Experimental design

As the authors rightly argued, with the changes made to the manuscript, this study can extend our understanding of the concomitance of these conditions and be a foundation for further studies aimed to reduce the burden of these diseases.

Validity of the findings

1. This study reviews the association of ADHD and AR in a number of rigorous studies and has potentially deepen our understanding of the concomitance of these conditions. Given the recent changes and kind discussion of the authors, the study is now appropriately fulfilling its objectives.
2. The revisions made by the authors seems to be adequate to advance the submission progress of this study.